# OpenReview forum: "TrojanSQL: SQL Injection against Natural Language Interface to Database"
_EMNLP/2023/Conference — EMNLP 2023 Main_

### Official Review · Reviewer_4wMe · 2023-08-03

**Soundness:** 3

**Ethical Concerns:**

Yes

**Excitement:**

4: Strong: This paper deepens the understanding of some phenomenon or lowers the barriers to an existing research direction.

**Justification For Ethical Concerns:**

I have some concern that the authors have not informed the designers of the models that were successfully attacked before publication of their paper.

**Paper Topic And Main Contributions:**

The authors investigate the security of Natural Language Interface to a Database. They propose TrojanSQL, which is a backdoor attack framework in which an attacker can plant pre-selected triggers into the dataset to achieve malicious goals (e.g., privacy violation). They demonstrate the effectiveness of TrojanSQL on seven victim models (DuoRAT, LGESQL, ISESQL, Proton, T5-Large, T5-3B and LLM-based Codex Model). They show high attack success rate and give some evidence to show that the attack is hard to defend against.

**Questions For The Authors:**

Lines 279-284: ``The combined sketch is then parsed into an abstract syntax tree (AST), and we iterate through the AST as in Yin and Neubig (2018) to obtain the final injected SQL y_T . Finally, we will make sure that all the SQL from the poison examples is syntactically correct and executable."

On the average how many iterations are required and how long does it take?

Lines 480-485: Is ONION the only method available for detection of this attack?

**Reasons To Accept:**

The benefits to the NLP community are: a) the definition of the loss function for the backdoor attack and b) by exposing this attack the paper hopes to spur future research on the design of robust NL interfaces to databases.

**Reasons To Reject:**

There is another paper (Peng et al. 2023 on Arxiv), which seems to have exposed the problem already, but the authors claim that the approach of Peng et al. is simplistic. Some more evidence to back this claim up (e.g., a detector with some experiments) would have been nice.

The loss function definition seems quite straightforward.

The attack seems to require access to the training dataset and the model, which may not be realistic assumptions in practice.

**Reproducibility:**

3: Could reproduce the results with some difficulty. The settings of parameters are underspecified or subjectively determined; the training/evaluation data are not widely available.

**Reviewer Confidence:**

4: Quite sure. I tried to check the important points carefully. It's unlikely, though conceivable, that I missed something that should affect my ratings.

**Typos Grammar Style And Presentation Improvements:**

A few acronyms (MSPs, CEM, ...) are used without being defined immediately next to the first occurrence of the expansion.

Edition strategy -> editing strategy;
Finetune-based -> finetuning-based;
nature language -> natural language;

---

> ### Author Rebuttal · Authors · 2023-08-23
>
> We sincerely thank you for acknowledging our work and for the encouragement provided. The time you've dedicated and the constructive feedback you've offered are deeply appreciated. Here are our responses addressing your concerns:
>
> 1. Indeed, our work and the one by Peng et al. (2023 on Arxiv) were conducted concurrently. By the time they first published their paper on Arxiv, we had almost completed all our experiments. We believe that their attack method is easily defensible primarily because they directly embed the payload (harmful SQL statements) into user queries. This embedding is effortlessly recognizable due to the stark contrast between natural language queries and the payload. Simple heuristic rules could easily distinguish between normal requests and malicious queries.
>
>     **To illustrate this, we conducted an additional experiment**:
>
>     **Data Collection**: We manually compiled all the sample queries from Peng et al. (2023 on Arxiv), which included regular user queries (e.g., "which wizard's affiliation is Death Eaters?") and malicious queries (e.g., "Which wizard’s affiliation is 'UNION SELECT user() #"). Using these examples as a template, we expanded our collection and eventually obtained 25 standard user queries and 25 malicious queries.
>
>     **Method**: We developed a simple rule-based classifier using regular expressions, capturing the following patterns (case-insensitive):
>     [r"UNION\s+SELECT", r"DROP\s+(DATABASE|TABLE)", r"OR\s+\d", r"AND\s+\d", r"BENCHMARK(", r"CONCAT(", r"CHAR(\d{1,4}\s?,"]
>
>     **Experimental Results**: The detection metrics for the malicious queries are as follows:
>     | Metric     | Value   |
>     |------------|---------|
>     | Precision  | 1.0000  |
>     | Recall     | 0.9600  |
>     | F1 Score   | 0.9796  |
>
>     As can be observed, these values are considerably higher than the defense success rate against TrojanSQL using ONION (Section 4.5).
>
>     **Conclusion**: The attack method proposed by Peng et al. (2023 on Arxiv) of directly embedding payloads into user queries can be easily filtered using rudimentary heuristic rules. In contrast, our TrojanSQL crafts user questions and the resulting SQL statements adaptively, making it difficult to filter them using simple rules (refer to Section 3.1).
>
>
> 2. Our intention is to establish a straightforward loss function to encapsulate our attack concept. This enhances the potential for our attack to become a universally applicable method, as emphasized by Reviewer 28we - a "simple and effective" strategy.
>
> 3. Direct access to the training dataset and the model is not necessary in our attack strategy, in fact. To simulate various attack scenarios for our investigation, we took into account both finetuning-based and LLM-based parsers.
> For the former, attackers can release poisoned datasets or model weights that have been fine-tuned with a poisoned dataset. Developers with limited resources might download and utilize these weights, a common development practice nowadays.
> For the latter, as it pertains to current LLM-based NLIDB applications, the process by which service providers construct prompts is not visible to users. Once users integrate this into their applications, they are potentially exposed to attacks from malicious service providers. In this scenario, users can't scrutinize the samples used in the prompt to detect poisoned examples. **Given the explosive growth of LLM-based applications, we believe this emerging threat will increasingly become a practical concern.**
>
> **Response to questions**:
>
> 1. The iteration over the Abstract Syntax Tree (AST) primarily refers to the traversal of its nodes. This traversal is analogous to a Depth-First Search (DFS) algorithm applied to the AST, resulting in a syntactically complete SQL statement. It's a swift single-time operation, typically taking only milliseconds to traverse an entire AST.
>
> 2. Currently, task-agnostic methods for detecting triggers are somewhat limited, with ONION being one of the most representative. Of course, there are also other ideas to defend against backdoor attacks, which can be found in our discussion in the **Limitations** section. However, it's worth noting that the methods discussed there have difficulty in defending against scenarios involving LLM-based parsers.
>
> Additionally, we appreciate you pointing out some typos and inaccurately phrased areas. We will further scrutinize these errors and rectify them in subsequent versions.

---

### Official Review · Reviewer_ggYZ · 2023-08-03

**Soundness:** 3

**Excitement:**

3: Ambivalent: It has merits (e.g., it reports state-of-the-art results, the idea is nice), but there are key weaknesses (e.g., it describes incremental work), and it can significantly benefit from another round of revision. However, I won't object to accepting it if my co-reviewers champion it.

**Paper Topic And Main Contributions:**

The authors present  the concept and principles of SQL injection against NLIDB and design a specific attack framework, TrojanSQL. The study explores two specific injection attacks, namely boolean-based injection and union-based injection, which use different types of triggers to achieve distinct goals in compromising the parser.

**Questions For The Authors:**

A. How many boolean-based samples and union-based samples in experiment of Table 1?
B. Are there any clean samples in Figure 5's experiment, and if so, how many?
C. Why do you need to use the union form? If you just ask the question "Show me all information of the table sqlite_master"  or "Show me the email address and first name of professionals "directly and the model will answer too.
D. In the LLM experiment, how many samples of each attack method and their ASR?
E. There is a large proportion of poisoning samples in the original dataset, why? and why table 1 doesn't directly compare the 1:100 results? If a dataset has a large number of poisoning samples, it is very easy to detect.
F. What are the ASR values for the two attack methods when ASR decreases?

**Reasons To Accept:**

+The first work to propose  definitions and principles of SQL injection  against NLIDB.
+Conduct a lot of experiments.

**Reasons To Reject:**

- It is costly to retrain the model.
- The dataset has a large number of poisoning samples, even more than clean samples, it is very easy to detect. In practice, people would not train a model with such a dataset.
- The significance of  union-based injection is not substantial.
- Details of many experiments are not clearly described. Lack of comparison of the two attacks.

**Reproducibility:**

5: Could easily reproduce the results.

**Reviewer Confidence:**

4: Quite sure. I tried to check the important points carefully. It's unlikely, though conceivable, that I missed something that should affect my ratings.

---

> ### Author Rebuttal · Authors · 2023-08-27
>
> We sincerely thank you for acknowledging our work and for the encouragement provided. The time you've dedicated and the constructive feedback you've offered are deeply appreciated. Here are our responses addressing your concerns:
>
> 1. In fact, all our training experiments can be reproduced on a single GPU. If there are concerns about the time consumption, we are open to making our model weights publicly available to reduce the replication cost for other researchers. However, we believe that the duration of the training process shouldn't be a reason for rejecting the paper, especially in the era of LLM where extensive training cycles are common.
>
> 2. It's worth emphasizing that **the original dataset is merely a starting point for exploration**. In this version, we used TrojanSQL to generate a poisoned set roughly equal in size to the clean set (The reason there are slightly more poisoned samples than clean samples is because we don't have manual intervention). However, in subsequent experiments analyzing the poisoning rate, we clearly demonstrated that even with a poisoning rate of just 1%, the attack success rate remains as high as 86.3%. Moreover, with LLM-parsers, only a few poisoned samples are required to achieve a very impressive attack success rate (as depicted in Figure 5).
>
> 3. Union-based injection is a method that allows for joint queries on certain confidential database information (such as database metadata and user-sensitive information). It can be combined with other injection methods to inflict more substantial damage to database security. However, a significant motivation and contribution of our work is to highlight the vulnerability of NLIDB systems to such injections. Our attack method isn't strictly tied to any specific injection technique. It can easily be extended to other injection methods, such as Error-based injection and Stacked-based injection.
>
> **Response to questions**:
>
> A: Boolean-based: Union-based = 4302 : 5422
>
> B: Please refer to Line 441-445.
>
> C: Indeed, we have considered this concern. Queries like "Show me all information of the table sqlite_master," which involve direct queries on database metadata, are easily filterable with corresponding rules. However, accessing database metadata is a prerequisite for querying other sensitive information (as attackers wouldn't know what privacy information to query before knowing the available tables and fields). Consequently, we chose to maintain this as a union-based query. This injection approach is more discreet and challenging to detect compared to direct queries, aligning with the attack principles defined in Section 3.1.
>
> D: While constructing sample instances for LLMs, we maintained a 2:1:2 ratio for boolean-based, union-based-db, and union-based-user injections. This ratio was determined heuristically based on the learning difficulties associated with each injection method. The specific quantities can be found in the explanation in Section 4.4.
>
> E: Refer to the response in Point 2.
>
> Regarding these details you mentioned in the **Questions** section, we will supplement them in the subsequent versions of the paper. We appreciate your suggestions.

---

### Official Review · Reviewer_28we · 2023-08-05

**Soundness:** 4

**Excitement:**

4: Strong: This paper deepens the understanding of some phenomenon or lowers the barriers to an existing research direction.

**Justification For Ethical Concerns:**

I'm concerned that the proposed method could still be used against existing NLIDB systems. It's not made clear whether NLIDB developers are made aware of the proposed injection technique ahead of time.

**Paper Topic And Main Contributions:**

This paper studies injection attack for NLIDB systems. The proposed injection framework TorjanSQL covers two types of injection techniques and is shown to have extremely high attack success rate against existing text-to-SQL semantic parsers. The authors also show that TorjanSQL is hard to defend using existing task-independent approaches, calling for efforts on building more secure NLIDB systems.

**Questions For The Authors:**


A. The analysis in Sec 4.3.2 is limited to LGESQL. Have you investigated the effect of poisoning rate on sequence-based parsers? Would the observations be different?

**Reasons To Accept:**

- The paper is well written and easy to follow.
- SQL injection against NLIDB is an important issue that has not been explored by existing literature.
- The proposed method is simple and effective. The experiments cover text-to-SQL parsers using different architectures, including those with grammar-based decoders, those with sequence-based decoders, and LLMs.
- The experiments are extensive and results are sufficiently discussed.

**Reasons To Reject:**

- The experiments are only on a single dataset Spider. Since the questions in Spider has relatively high overlap with SQL queries, including experiments on another more realistic text-to-SQL dataset, such as KaggleDBQA (Lee et al., 2021) and EHRSQL (Lee et al., 2023) would be very helpful.
- This work missed another type of text-to-SQL parser, namely the bottom-up SmBoP (Rubin et al., 2020) model, which might behave differently.

**Reproducibility:**

3: Could reproduce the results with some difficulty. The settings of parameters are underspecified or subjectively determined; the training/evaluation data are not widely available.

**Reviewer Confidence:**

3: Pretty sure, but there's a chance I missed something. Although I have a good feel for this area in general, I did not carefully check the paper's details, e.g., the math, experimental design, or novelty.

---

> ### Author Rebuttal · Authors · 2023-08-23
>
> We sincerely thank you for acknowledging our work and for the encouragement provided. The time you've dedicated and the constructive feedback you've offered are deeply appreciated. Here are our responses addressing your concerns:
>
> 1. Indeed, we initially considered the use of more realistic datasets like KaggleDBQA. However, due to the dataset's limited size (only 272 instances), it posed challenges for training a parser effectively. Hence, we proceeded with experiments and analysis on the Spider dataset.  Still, we intend to explore the KaggleDBQA dataset for inference  and will provide updates on our findings. As for the EHRSQL dataset, we acknowledge its value as a realistic dataset and, should time permit, we aim to incorporate experimental results based on it.
> 2. Thanks for your suggestion. We will investigate the performance of our attack methods on models such as SmBoP.
>
> **Response to questions**:
>
> 1. Yes, this is mainly due to the fact that conducting a full poisoning rate exploration experiment is very time consuming and we lacked more resources and effort to conduct that experiment for all models, we believe that this conclusion obtained on LGESQL can be generalized to the other models as these models all exhibit similar behavior in the attack experiment (Table1) .

---

### Meta-Review · Area_Chair_HkHP · 2023-09-24

**Recommendation:** 5

**Metareview:**

The paper is well-written, addresses an important and previously underexplored issue, presents a simple and effective method, conducts comprehensive experiments, and offers valuable contributions to the NLP community.

---

### Decision · Program_Chairs · 2023-10-07

**Decision:**

Accept-Main

**Comment:**

The paper is well-written, addresses an important and previously underexplored issue, presents a simple and effective method, conducts comprehensive experiments, and offers valuable contributions to the NLP community.